# Contact Tracing and Tuberculosis Preventive Therapy for Household Child Contacts of Pulmonary Tuberculosis Patients in the Kyrgyz Republic: How Well Are We Doing?

**DOI:** 10.3390/tropicalmed8070332

**Published:** 2023-06-21

**Authors:** Meder Kadyrov, Pruthu Thekkur, Evgenia Geliukh, Aelita Sargsyan, Olga Goncharova, Aizat Kulzhabaeva, Asel Kadyrov, Mohammed Khogali, Anthony D. Harries, Abdullaat Kadyrov

**Affiliations:** 1National Centre of Phthisiology, Ministry of Health, Bishkek 720000, Kyrgyzstan; goncharova.ncph@gmail.com (O.G.); abdylat.kadyrov@gmail.com (A.K.); 2Centre for Operational Research, International Union Against Tuberculosis and Lung Disease, 2 Rue Jean Lantier, 75001 Paris, France; pruthu.tk@theunion.org (P.T.); adharries@theunion.org (A.D.H.); 3International Charitable Foundation “Alliance for Public Health”, 01601 Kiev, Ukraine; geliukh73@gmail.com; 4Tuberculosis Research and Prevention Centre (TBRPC), Yerevan 0014, Armenia; sargsyan.aelita@gmail.com; 5Public Foundation KNCV-KG, Bishkek 720000, Kyrgyzstan; a.kulzhabaeva@list.ru; 6Primary Healthcare Centre #1, Bishkek 720000, Kyrgyzstan; kadyrova.asel.74@gmail.com; 7Institute of Public Health (IPH), College of Medicine and health Sciences (CMHS), United Arab Emirates University (UAEU), Al Ain 15551, United Arab Emirates; ahmedm@uaeu.ac.ae; 8Department of Clinical Research, Faculty of Infectious and Tropical Diseases, London School of Hygiene and Tropical Medicine, London WC1E 7HT, UK

**Keywords:** contact investigation, household contact management, childhood TB, pediatric TB, isoniazid preventive therapy, structured operational research training initiative (SORT IT), operational research

## Abstract

Early identification, screening and investigation for tuberculosis (TB), and provision of TB preventive therapy (TPT), reduces risk of TB among child household contacts of pulmonary TB patients (index patients). A cohort study was conducted to describe the care cascade and timeliness of contact tracing and TPT initiation among child household contacts (aged < 15 years) of index patients initiated on TB treatment in Bishkek, the Kyrgyz Republic during October 2021–September 2022. In the register, information on the number of child household contacts was available for 153 (18%) of 873 index patients. Of 297 child household contacts identified, data were available for 285, of whom 261 (92%) were screened for TB. More than 50% were screened after 1 month of the index patient initiating TB treatment. TB was diagnosed in 23/285 (9%, 95% CI: 6–13%) children. Of 238 TB-free children, 130 (55%) were eligible for TPT. Of the latter, 64 (49%) were initiated on TPT, of whom 52 (81%) completed TPT. While TPT completion was excellent, there was deficiency in contact identification, timely screening and TPT initiation. Thus, healthcare providers should diligently request and record details of child household contacts, adhere to contact tracing timelines and counsel caregivers regarding TPT.

## 1. Introduction

Tuberculosis (TB) is a global public health problem. It is estimated that 10.6 million individuals fell ill and 1.6 million died due to TB in 2021 [1]. Children aged less than 15 years account for 11% (about 1.2 million) of the estimated TB patients [1]. However, most children with TB remain undetected with a 22% risk of death due to TB, which can reach up to 44% in children under 5 years of age [1,2,3]. 

TB infection is acquired by inhalation of infectious droplets with Mycobacterium tuberculosis bacilli generated from coughing or sneezing by patients with active infectious pulmonary TB disease. Inhaled bacilli reach lung alveoli and get encapsulated in the granuloma due to immune response without TB disease and being infectious to other people. However, when conducive, the bacilli escape the immune system to multiply and cause active infectious pulmonary TB disease. 

Children mainly contract TB infection after close contact with an active pulmonary TB patient (index TB patient) at the household level [4]. Children with TB infection have a 10% risk of developing TB disease in their lifetime, with 83% developing the disease within the first 3 months of exposure [4,5]. A meta-analysis showed that timely provision of TB preventive therapy (TPT) could reduce the risk of developing TB disease by 63% among all children exposed to active pulmonary TB patients and by 85% among those identified with TB infection [4]. Therefore, the World Health Organization (WHO) recommends contact tracing (screening) of all child household contacts of pulmonary TB patients (index TB patient) and early initiation of anti-TB treatment for those diagnosed with TB disease and provision of TPT for those without TB [6]. 

Even though TPT is one of the key interventions under patient-centered care and prevention of the WHO’s End TB Strategy, only 32% of child household contacts (aged <5 years) received TPT in 2021 globally [1,7]. Additionally, during the United Nations High-Level Meeting on the fight against TB in 2018, world leaders committed to providing TPT to 4 million household contacts aged < 5 years and 20 million household contacts aged ≥ 5 years by 2022 [8]. Despite the high political commitment made by member states, only 1.6 million household contacts aged < 5 years (40% of the target) and 0.6 million aged ≥ 5 years (3% of the target) had received TPT by 2021 [1]. Studies from low-and-middle-income countries (LMICs) have reported sup-optimal implementation of contact tracing and highlighted reluctance among providers and parents to provide TPT [9,10,11,12,13,14,15,16].

The Kyrgyz Republic is a low-income country in Central Asia with high multi-drug/rifampicin resistance TB (MDR/RR TB) burden [1]. In the Kyrgyz Republic, only 52% of the estimated TB patients were detected in 2021 and only 5% of those detected were children [1]. This highlights the under-detection of TB in the country, especially among children. Furthermore, only 11% of child household contacts of TB patients (aged < 5 years) were started on TPT in 2021, well below the global TPT coverage (32%) [1]. This low coverage of TPT is a concern taking into consideration the relatively high TB infection rate (33%) among child household contacts in the country [17]. 

Since 1970, the National TB Programme (NTP) in the Kyrgyz Republic has been implementing household contact tracing and the provision of TPT. Contact tracing and provision of TPT includes sequential steps (cascade), starting with the epidemiological investigation of the index TB patient, household contact screening for symptoms suggestive of TB, contact clinical examination to rule out active TB, ascertainment of eligibility for TPT and initiation of TPT. Consequently, the NTP of the Kyrgyz republic felt it is necessary to assess the performance of the contact tracing and provision of TPT cascade. Such information will help understand the coverage and timeliness of each step of the cascade, identify gaps and make informed decisions to improve performance. Additionally, this information will provide baseline data for future monitoring of the performance of contact tracing and provision of TPT.

Thus, among child household contacts (aged < 15 years) of pulmonary TB patients initiated on anti-TB treatment from October 2021 to September 2022 in Bishkek, the Kyrgyz Republic, we aimed to describe the care cascade and timeliness of contact tracing and TPT provision. Specific objectives were to assess: (1) the number and proportion of child household contacts who were screened for TB, underwent TB diagnostic tests, were diagnosed with TB and initiated on anti-TB treatment; (2) the number and proportion who were eligible for TPT, initiated and completed TPT and (3) the median (inter-quartile range) number of days taken to complete each step of the cascade. 

## 2. Materials and Methods

### 2.1. Study Design

We conducted a cohort study using secondary data routinely collected by the Bishkek Tuberculosis Centre, functioning under the NTP of the Kyrgyz Republic.

### 2.2. Study Setting

#### 2.2.1. General Setting

The Kyrgyz Republic is a landlocked country, sharing borders with Kazakhstan, Tajikistan, Uzbekistan and China. It has a population of about 6.7 million, of whom 17% are children aged less than 15 years [18]. The country is divided administratively into seven regions (Oblast) and two large cities of republican importance—Bishkek and Osh. A quarter of the country’s population (25%) lives in relative poverty [19]. The Ministry of Health regulates the delivery of health services in the country through its three-tier public healthcare system. 

#### 2.2.2. Specific Setting

This study was conducted in Bishkek, which is the capital city of the Kyrgyz Republic. The city has about 1 million inhabitants, with about 30% of them being children aged less than 15 years [18]. The city is administratively divided into four districts: Pervomay, Leninsky, Oktyabrsky and Sverdlovsky. Healthcare services are provided through public and private health facilities spread across the city. The city has 10 primary healthcare centers (PHCs), also called polyclinics, providing preventive and curative services to people from a defined geographic catchment area. There are also secondary and tertiary public health facilities, including disease specific hospitals like the Bishkek Tuberculosis Centre.

##### Diagnosis and Treatment of Index TB Patient in Bishkek

The individuals presenting with symptoms suggestive of TB to the PHCs and other public health facilities are identified as presumptive TB patients and referred to the Bishkek Tuberculosis Centre for confirmation of diagnosis and treatment initiation. Patients diagnosed by private providers are also referred to the Bishkek Tuberculosis Centre for treatment initiation. At treatment initiation, the TB patients are registered and their demographic and clinical details are recorded in the TB register (Journal TB-02) maintained in the Bishkek Tuberculosis Centre. The aggregate number of child household contacts aged less than 15 years for each patient is also recorded in the register. A structured medical card is maintained for each patient and has provision for recording contact tracing information of all child household contacts.

##### Contact Tracing of Pulmonary TB Patients

(a)Identification of household child TB contacts:

After initiating anti-TB treatment in a pulmonary TB patient (including for drug-resistant and clinically diagnosed disease) at Bishkek Tuberculosis Centre, the TB physician provides the patient’s contact details to the epidemiologist at the Sanitary Epidemiological Service unit. The epidemiologist then conducts the epidemiological investigation of the index patient and lists all the household contacts, and additionally, informs the household contacts to visit the assigned PHC for further screening for TB and TPT provision. The epidemiologist then shares the list of all the contacts with the family physician in the PHC and alerts them about the need to conduct screening and contact examination of child household contacts. 

(b)Contact examination of household child contacts:

The family doctor at the PHC is responsible for conducting the screening and contact examination of all the child household contacts. All child household contacts (aged < 15 years) are requested to visit the facility or the Bishkek Tuberculosis Centre to be screened for symptoms suggestive of TB, such as cough for over two weeks, hemoptysis, weight loss, fever, night sweats and decreased appetite. Additionally, the child household contacts undergo a chest X-ray. Those with any symptoms suggestive of TB or with an abnormal chest X-ray are considered to have presumptive TB and are evaluated for TB with sputum smear microscopy or Xpert MTB/RIF assay. For child household contacts who are symptom-free and aged 5 to 14 years, a tuberculin skin test (TST) is performed at PHCs. After screening and examination, all the child household contacts are referred to physicians at the Bishkek Tuberculosis Centre for making decisions on TB diagnosis and TPT. It is recommended that contact tracing is completed and decisions are taken within 7 to 14 days.

(c)Diagnosis of TB and TPT eligibility ascertainment:

The TB physician assesses all child household contacts referred to the Bishkek Tuberculosis Centre. The child household contacts with either bacteriologically confirmed or clinically diagnosed TB are initiated on anti-TB treatment regimen based on the drug-sensitivity pattern. All child household contacts who do not have TB are assessed for TPT eligibility. The national guidelines recommend TPT for child household contacts of drug-sensitive bacteriologically confirmed pulmonary TB patients who are: aged under 5 years;aged 5 to 14 years and have positive TST results.

However, in practice, the child household contacts of some drug-resistant pulmonary TB patients are also considered eligible for TPT due to uncertainty about drug resistance status and delays in drug susceptibility test results of index TB patients. After treatment initiation, the “bacteriological confirmation” and “drug-resistant status” of index TB patients could change based on the results of further investigations like Xpert MTB/RIF assay or mycobacterial culture. However, the details remain unchanged in the TB Journal-02 register.

(d)TPT initiation and follow-up: 

All child household contacts eligible for TPT are initiated on a six-month regimen of isoniazid monotherapy (even for drug-resistant TB contacts) at the Bishkek Tuberculosis Centre. Each child contact initiated on TPT at Bishkek Tuberculosis Centre is issued a medical card for recording information on TPT provision and follow-up. The contacts are referred back to the PHC for receipt of TPT and follow-up. TPT drugs are issued weekly from the PHC and administered under the caregiver’s supervision. TPT is stopped and anti-TB treatment is initiated among those who develop active TB. Among those aged 5 to 14 years, repeat TST is conducted at 2 months; if TST is negative, TPT is considered completed and stopped.

### 2.3. Study Population

All pulmonary TB patients (including those with drug-resistant and clinically diagnosed disease) initiated on anti-TB treatment during the period from October 2021 to September 2022 in Bishkek, Kyrgyzstan and their child household contacts aged less than 15 years were included in the study. 

### 2.4. Data Collection, Variables and Study Tools

The research team collected the data for the study in April 2023. All the data variables required for the study were extracted from the Journal TB-02, the medical card of the index TB patient and the contact medical card maintained at the Bishkek Tuberculosis Centre. The demographic details of index TB patients, such as age, gender, district, type of case, type of TB, drug resistance status and number of child household contacts aged less than 15 years, were extracted from the Journal TB-02. The details of the child household contacts—such as age, gender, relationship with index TB patient, screening for symptoms, date of screening, chest X-ray findings, diagnosis of TB, type of TB, date of initiating anti-TB treatment, TST results, TPT initiation, date of TPT initiation, TPT completion and date of TPT completion—were extracted from the medical records of the index TB patient. Using the extracted variables, we also derived variables like child household contact with “presumptive TB” and “eligible for TPT”. The operational definitions used for the derived variables are mentioned below.

#### Operational Definition

Screened for TPT: All child household contacts for whom there was an indication of screening for symptoms suggestive of TB in the medical card of the index TB patients or contact medical record.

Presumptive TB: All child household contacts with either symptom suggestive of TB or an abnormal chest X-ray.

Eligible for TPT: In this study, this refers to all TB-free child household contacts aged less than 5 years and child household contacts aged 5 to 15 years with positive TST (no details on the exact size of induration). 

TPT completed: All child household contacts for whom there is an indication of completion of TPT in the medical card of index TB patients or contact medical record. 

### 2.5. Data Analysis

The extracted data were directly entered into a Microsoft Excel workbook. Data were analyzed using STATA^®^ (version 16.0 Copyright 1985–2019 StataCorp LLC, College Station, TX, USA). The demographic and clinical characteristics of household child TB contacts and their index TB patients were summarized using frequencies and percentages.

The number and percentages with 95% confidence intervals (CI) were calculated for the key metrics of the care cascade, such as (a) screened for symptoms of TB among all child household contacts, (b) diagnosed with TB among those screened and (c) eligible for TPT among those without TB and initiated on TPT among those eligible for TPT. The chi-square test was used to compare the frequency and percentage of key metrics across demographic and clinical characteristics of child household contacts and index TB patients. The level of statistical significance was set at *p* ≤ 0.05 for all statistical tests.

Median and inter-quartile range (IQR) was used to summarize the duration between treatment initiation of the index patient and contact screening, the duration between contact screening to initiation of anti-TB treatment among those initiated on anti-TB treatment and the duration between contact screening to initiation of TB preventive therapy among those initiated on TPT. We also calculated the frequency and percentage of child household contacts initiated on anti-TB treatment or TPT within 15 days of initiating anti-TB treatment in the index patient. 

## 3. Results

We included 873 pulmonary TB patients (index TB patients) initiated on anti-TB treatment during the study reference period in Bishkek and their TB Journal-02 details were extracted. In 153 (18%) of 873 index TB patients, there was a mention of at least one child household contact in the field “number of household contact < 15 years”; for the remainder, the field was blank in the TB Journal-02. There were a total of 297 child household contacts as per this field. Of these, 285 (96%) child household contacts for whom individual level data were available in the medical card of the index patient or the contact medical card were included for further analysis (Figure 1).

### 3.1. Demographic and Clinical Details of Child Household Contact (Including That of Index TB Patients)

Of the 285 child household contacts, 173 (61%) were aged between 5 to 14 years, 151 (53%) were males and 155 (54%) were children of the index TB patients. The majority of the child household contacts were contacts of the index TB patients who were registered as bacteriologically confirmed TB (83%) and drug-sensitive TB (70%) at the time of treatment initiation. Of the total, 58 (20%) were contacts of MDR/RR/XDR TB patients (Table 1).

### 3.2. Screening and Diagnosis of TB

Of the 285 child household contacts, 261 (92%, 95% CI: 88–95%) were screened for symptoms of TB, of whom 13 (5%) had symptoms suggestive of TB. Of those screened for symptoms, 260 (~100%) underwent chest X-ray and 24 (9%) had abnormal X-ray findings. Of 261 child household contacts screened, 25 (10%, 95% CI: 6–14%) had presumptive TB and 23 (9%, 95% CI: 6–13%) were diagnosed with TB (Figure 1). 

The proportion screened for TB symptoms was significantly lower among child household contacts from Leninsky district (85%) and among those who had an “other” relationship to the index patient (71%) compared with those who were a child, a grandchild or a sibling of the index patient. A significantly higher proportion of child household contacts of retreatment index TB patients (14%) were diagnosed with TB (Table 2).

### 3.3. Eligibility for TPT, TPT Initiation and Completion

Of the 238 child household contacts who were TB free, 92 were aged < 5 years and were eligible for TPT without further investigation. Of the 146 child household contacts aged 5 to 14 years, 144 (99%) underwent TST, of whom 38 (26%) were TST-positive and eligible for TPT. Thus, 130 (55%, 95% CI: 48–61%) of the total 238 child household contacts who were TB free were eligible TPT.

Of the 130 eligible for TPT, 64 (49%, 95% CI: 40–58%) were initiated on TPT, of whom 52 (81%, 95% CI: 70–90%) completed TPT (Figure 1). The proportion initiated on TPT was significantly lower among child household contacts aged < 5 years compared to those who were 5 to 14 years. The TPT completion rate was significantly lower in Pervomay (55%) district (Table 3). 

### 3.4. The Time Taken from Index Patient Treatment Initiation to Screening and Initiation of Anti-TB Treatment or TPT

Of the 261 child household contacts who were screened for symptoms suggestive of TB, only 200 (77%) had a valid date of symptom screening. Among 200 child household contacts, the median duration from treatment initiation of the index patient to screening of contacts was 45 days (IQR 8–100 days). After symptom screening, the median duration to start anti-TB treatment or TPT was 24 days (IQR 6–74 days). There was no statistically significant difference in the time taken to complete various steps between child household contacts aged under 5 years and those who were 5 to 14 years. Only 29 (35%) of 82 child household contacts on anti-TB treatment or TPT were initiated on treatment within 15 days of initiating treatment for the index TB patient (Table 4).

## 4. Discussion

This is the first study from the Kyrgyz Republic that has assessed the cascade of contact tracing and provision of TPT for child household contacts (aged < 15 years) of pulmonary TB patients. The study has five key findings of programmatic importance. First, the child household contacts were listed in only one out of five pulmonary TB patients. Second, nine out of ten child household contacts listed were screened for TB and underwent investigations to diagnose TB and assess TPT eligibility. Third, over half of the child household contacts were screened after six weeks of the index patient initiating anti-TB treatment. Fourth, only half of the child household contacts eligible for TPT were initiated on TPT. Finally, more than two-thirds of those initiated on TPT completed the treatment.

The study had several strengths. First, the study findings reflect the programmatic realities of contact tracing and TPT provision as we used data routinely collected by the NTP. Second, there was less scope for selection bias as all the pulmonary TB patients who initiated anti-TB treatment in Bishkek (the largest city of Kyrgyz Republic) during the study reference period were included. Third, the study is in line with the research priority of the NTP, and thus, has potential for research uptake for making informed decisions to improve household contact tracing and TPT provision. Fourth, we also adhered to the STROBE (Strengthening the Reporting of Observational Studies in Epidemiology) guidelines for conducting and reporting observational studies [20].

The study had some important limitations. First, due to the relatively small sample of child household contacts in the study, the estimates of TPT initiation and completion were not precise. Additionally, we could not perform an adjusted analysis for assessing independent child and index patient characteristics associated with gaps in each step of the cascade. Second, we included index patients and their contacts only from the capital city Bishkek, and they might have had better literacy, economic status and access to healthcare services. Thus, the study findings might not be generalizable to the whole of the Kyrgyz Republic. Third, there were deficiencies (missing and implausible data) in the data sources that were used for the study. The field “number of household contacts < 15 years” was left blank in the TB Journal-02 for 82% of index TB patients. Other key date variables required for the study were missing in the medical cards of index patients and contacts maintained in the Bishkek Tuberculosis Centre. There were also inconsistencies in the dates recorded for the sequential steps of contact tracing. Finally, we included all the pulmonary TB patients, though the national guidelines recommend TPT for eligible child household contacts of only bacteriologically confirmed drug-sensitive pulmonary TB patients. We had to include all pulmonary TB patients (not limiting to bacteriologically confirmed drug-sensitive pulmonary TB patients) due to the potential misclassification of “bacteriological confirmation” and “drug-resistant status” of index TB patients in the Journal TB-02 filled at the time of treatment initiation. However, including all pulmonary TB patients would have contributed to overestimation of the child household contacts eligible for TPT and underestimation of TPT initiation. 

There are several policy and practice implications from the study findings. First, the child household contacts were listed only for 1 out of 5 (18%) index TB patients in a country that is home to 30% of the population aged < 15 years [18]. A pilot project from the Kyrgyz Republic on household contact tracing through home visits had reported 111 child household contacts for 67 index TB patients, way higher than 297 listed for 873 index TB patients in the current study [17]. In addition, studies from Armenia (74%), South Africa (46%), Tajikistan (37%), India (30%) and Ethiopia (30%) have reported higher percentages child household contacts amongst index TB patients [21,22,23,24,25]. Identification of the child household contacts of index TB patients is a critical step in the cascade and deficiencies here, and furthermore, could result in the missing of child household contacts in the early stages of contact tracing [9,26]. In the current context, we are not sure whether the healthcare providers failed to ask index TB patients about their household contacts or whether they asked but did not record it under the field “the number of household contacts < 15 years” in Journal TB-02. There is a need for further qualitative exploration to understand the reason for this gap. However, it should be re-emphasized to the healthcare providers that they should ask all index TB patients about their household contacts < 15 years and record this in the register. In case the index patient has no child household contact, the healthcare providers should record “zero” under the field “number of household contacts < 15 years”. The monitoring team should periodically assess the completeness of the recording of contact identification. 

Second, although a high percentage (90%) of child household contacts listed were screened for TB, more than 50% were screened after 6 weeks of the index patient initiating anti-TB treatment. This sort of delay in contact screening would eventually delay the anti-TB treatment or TPT for the eligible child household contacts. As child household contacts have a relatively high risk of developing TB disease within three months of exposure to index TB patients, delay in TPT provision would be a waste of the opportunity to prevent TB disease [4,5]. Thus, the NTP should consider introducing timelines for each step of the care cascade to ensure timely TPT provision. The NTP can also institute and monitor metrics like “the number and proportion of child household contacts completing each step within the recommended timelines”. 

Third, almost all child household contacts who were screened underwent investigations like a chest X-ray and TST for TB diagnosis and TPT eligibility assessment. However, studies from other countries have reported a low uptake of these investigations [5,13,24,25]. One of the possible reasons for better uptake of investigations is the availability of chest X-ray, TST and physician consultation at the Bishkek Tuberculosis Centre. Thus, child household contacts in Bishkek do not need to visit multiple facilities and all the investigations and the consultation can be done in a single visit. This highlights the importance of developing one-stop facilities for child household contact investigation for minimizing lost-to-follow up, ensuring better uptake of investigations and facilitating early decisions on further management [9]. 

Fourth, only 49% of the eligible child household contacts were initiated on TPT. However, of those initiated on TPT, the majority (83%) completed treatment. This finding of low TPT initiation rate and high completion rate is in line with the evidence from other countries [10,11,12,13,14,22,25,27,28]. Though the TPT initiation rate was relatively low (33%) among child household contacts aged < 5 years, it was 3 times higher than that reported from the Kyrgyz Republic to the WHO in 2021 [1]. This means the TPT initiation rate could be poorer in other parts of the country compared to Bishkek city, which has better health facilities. Therefore, there is an urgent need to engage with TB physicians and healthcare providers to understand the contextual barriers to TPT initiation and implement evidence-based strategies to overcome barriers, including providing a shorter TPT regimen [9]. Moreover, there is a need to educate caregivers, index TB patients and the community about the importance of TPT [9].

Fifth, although not recommended in the national guidelines, child household contacts of drug-resistant TB patients (seven contacts) were initiated on six-months isoniazid therapy. This could be due to misclassification of drug-resistant TB at the time of registration. However, the provision of six months of isoniazid to those detected as drug-resistant TB based on the Xpert MTB/RIF assay or mycobacterial culture and drug-susceptibility test results is a concern. The WHO recommends a fluoroquinolone-based regimen for contacts of drug-resistant TB patients [6]. As the country has a high burden of drug-resistant TB [1], there is a need to develop country-specific guidelines on TPT regimens for child household contacts of drug-resistant TB patients. 

Sixth, several deficiencies identified in the recording of contact tracing and TPT provision need urgent attention. This underlines the importance of the NTP’s current efforts to digitize TB patients’ medical cards to improve the TB recording and reporting system. In the electronic records, provision should be made for recording details of child household contact identification, contact investigation and TPT provision. Moving forward, dashboards should be built using electronic medical cards for real-time monitoring of contact tracing performance and data-informed decisions for improving the process.

## 5. Conclusions

This first study from the Kyrgyz Republic on contact tracing and TPT provision showed that the child household contacts were identified only in one out of five pulmonary TB patients, it took more than one month to screen the majority of child household contacts after treatment initiation of the index TB patient and only half of the eligible child household contacts were initiated on TPT. Therefore, there is a need for further qualitative exploration to understand the reason for these gaps and implement evidence-based strategies to improve programme performance. As an immediate response, healthcare providers should be trained on the importance and the need to: (1) request and list all the child household contacts of pulmonary TB patients at the time of initiating anti-TB treatment; (2) educate caregivers and index TB patients about the importance of TPT and convince them to initiate TPT for the child household contacts and (3) improve recording of contact tracing and TPT provision. 

## Figures and Tables

**Figure 1 tropicalmed-08-00332-f001:**
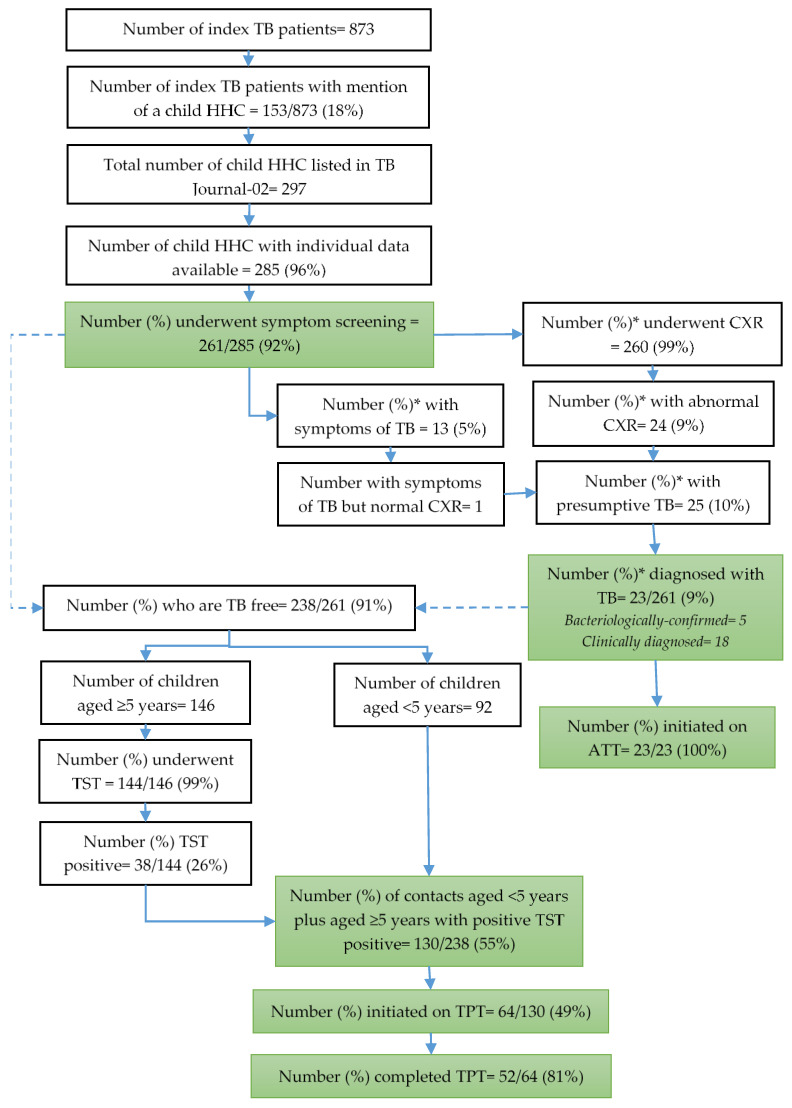
Care cascade of contact tracing and TPT provision for child household contacts of pulmonary TB patients initiated on anti-TB treatment in Bishkek, Kyrgyzstan, during 2021–2022. * Percentage calculated with number undergoing symptom screening as the denominator (N = 261). Abbreviation: TB = Tuberculosis; CXR = Chest X-ray, SSM = Sputum Smear Microscopy; TST = Tuberculin Skin Test; ATT = Anti-TB treatment; TPT = TB preventive therapy.

**Table 1 tropicalmed-08-00332-t001:** Demographic and clinical characteristics (index patients) of the child household contacts of pulmonary TB patients initiated on anti-TB treatment in Bishkek, Kyrgyzstan, during October 2021 to September 2022.

Characteristics	Total	
	n	(%) ^1^
Total	285	
*Child contact characteristics*		
Age (in completed years)		
0–4	112	(39.3)
5–14	173	(60.7)
Gender		
Male	151	(53.0)
Female	130	(45.6)
Not recorded	4	(1.4)
Relationship with Index patient		
Child	155	(54.4)
Grandchild	49	(17.2)
Sibling	56	(19.7)
Others	24	(8.4)
Not recorded	1	(0.4)
Districts		
Pervomay	119	41.8
Leninsky	110	38.6
Oktyabrsky	10	3.5
Sverdlovsky	46	16.1
*Index patient characteristics*		
Type of Case		
New	137	(48.1)
Retreatment	140	(49.1)
Not recorded	8	(2.8)
Type of TB		
Bacteriologically confirmed	236	(82.8)
Clinically diagnosed	41	(14.4)
Not recorded	8	(2.8)
Drug resistance		
Sensitive	199	(69.8)
RR/MDR/XDR	58	(20.4)
Poly resistant	20	(7.0)
Not recorded	8	(2.8)

^1^ Column percentage; Abbreviation: TB = Tuberculosis; RR = Rifampicin resistant; MDR = Multi-drug resistant; XDR = Extensively Drug-Resistant.

**Table 2 tropicalmed-08-00332-t002:** Number and proportion screened for symptoms of TB and diagnosed with TB stratified by the demographic and clinical characteristics of the child household contacts of pulmonary TB patients initiated on the anti-TB treatment in Bishkek, Kyrgyzstan during October 2021 to September 2022.

Characteristics	Total	Screened ^1^		*p*-Value ^3^	Diagnosed TB		*p*-Value ^3^
	(a)	n	(%) ^2^		n	(%) ^4^	
Total	285	261	(91.6)		23	(8.8)	
*Child contact characteristics*							
Age in years							
<5	112	103	(92.0)	0.850	11	(10.5)	0.155
5–14	173	158	(91.3)		12	(7.6)	
Gender							
Male	151	136	(90.1)	<0.001	10	(7.4)	0.641
Female	130	124	(95.4)		13	(10.5)	
Not recorded	4	1	(25.0)		0	(0.0)	
Relationship with Index patient							
Child	155	148	(95.5)	<0.001	13	(8.8)	0.083
Grandchild	49	41	(83.7)		0	(0)	
Sibling	56	55	(98.2)		7	(12.7)	
Others	24	17	(70.8)		3	(17.7)	
Not recorded	1	0	(0)				
Districts							
Pervomay	119	116	(97.5)	0.005	9	(7.8)	0.484
Leninsky	110	93	(84.6)		8	(8.6)	
Oktyabrsky	10	9	(90.0)		0	(0)	
Sverdlovsky	46	43	(93.5)		6	(14.0)	
*Index patient characteristics*							
Type of TB							
New	137	129	(94.2)	0.168	5	(3.9)	0.021
Retreatment	140	124	(88.6)		17	(13.7)	
Not recorded	8	8	(100)		1	(12.5)	
Type of diagnosis							
Bacteriologically confirmed	236	217	(92.0)	0.464	18	(8.3)	0.801
Clinically diagnosed	41	36	(87.8)		4	(11.1)	
Not recorded	8	8	(100)		1	(12.5)	
Drug resistance							
Sensitive	199	181	(91.0)	0.103	17	(9.4)	0.624
RR/MDR/XDR	58	56	(96.6)		5	(8.9)	
Poly resistant	20	16	(80.0)		0	(0)	
Not recorded	8	8	(100)		1	(12.5)	

^1^ Screened for symptoms of TB; ^2^ Row percentage with total number of children in each category as a denominator; ^3^
*p* value derived with chi-square test; ^4^ Row percentage with the number of screened in each category as the denominator; Abbreviation: TB = Tuberculosis; RR = Rifampicin resistant; MDR = Multi-drug resistant; XDR = Extensively Drug-Resistant.

**Table 3 tropicalmed-08-00332-t003:** Number and proportion eligible for TPT, initiated and completed TPT stratified by the demographic and clinical characteristics of the child household contacts of pulmonary TB patients initiated on the anti-TB treatment in Bishkek, Kyrgyzstan during October 2021 to September 2022.

Characteristics	Total	Eligible for TPT ^1^		*p* Value ^3^	Initiated on TPT		*p* Value ^3^	Completed TPT		*p* Value ^3^
		n	(%) ^2^		n	(%) ^4^		n	(%) ^5^	
Total	238	130	(54.6)		64	(49.2)		52	(82.5)	
*Child contact characteristics*										
Age (in completed years)										
<5	92	92	(100)	<0.001	31	(33.7)	<0.001	22	(73.3)	0.066
5–14	146	38	(26.0)		33	(86.8)		30	(90.9)	
Gender										
Male	126	66	(52.4)	0.525	33	(50.0)	0.573	26	(81.3)	0.877
Female	111	63	(56.8)		30	(47.6)		23	(83.3)	
Not recorded	1	1	(100)		1	(100)		1	(100)	
Relationship with index patient										
Child	135	76	(56.3)	0.198	33	(43.4)	0.013	28	(87.5)	0.565
Grandchild	41	17	(41.5)		5	(29.4)		4	(80.0)	
Sibling	48	27	(56.3)		18	(66.7)		13	(72.2)	
Others	14	10	(71.4)		8	(80.0)		7	(87.5)	
Districts										
Pervomay	107	49	(45.8)	0.052	22	(44.9)	0.085	12	(54.6)	<0.001
Leninsky	85	56	(65.9)		32	(57.1)		30	(96.8)	
Oktyabrsky	9	5	(55.6)		0	(0)				
Sverdlovsky	37	20	(54.1)		10	(50.0)		10	(100)	
*Index patient characteristics*										
Type of TB										
New	124	76	(61.3)	0.085	38	(50.0)	0.525	33	(89.2)	0.108
Retreatment	107	50	(46.7)		23	(46.0)		16	(69.6)	
Not recorded	7	4	(57.1)		3	(75.0)		3	(100)	
Type of diagnosis										
Bacteriologically confirmed	199	107	(53.8)	0.832	53	(49.5)	0.484	45	(86.5)	0.029
Clinically diagnosed	32	19	(59.4)		8	(42.1)		4	(50.0)	
Not recorded	7	4	(57.1)		3	(75.0)		3	(100)	
Resistance										
Sensitive	164	93	(56.7)	0.780	47	(50.5)	0.058	35	(76.1)	0.177
RR/MDR/XDR	51	25	(49.0)		7	(28.0)		7	(100)	
Poly resistant	16	8	(50.0)		7	(87.5)		7	(100)	
Not recorded	7	4	(57.1)		64	(49.2)		3	(100)	

^1^ Those under 5 years of age and those aged 5–14 years with positive TST test were considered eligible for TPT; ^2^ Row percentage with total number of children who were TB free in each category as a denominator; ^3^
*p* value derived with chi-square test; ^4^ Row percentage with the number eligible for TPT in each category as the denominator; ^5^ Row percentage with the number initiated on TPT in each category as the denominator; Abbreviation: TB = Tuberculosis; RR = Rifampicin Resistant; MDR = Multidrug Resistant; XDR = Extensively Drug-Resistant.

**Table 4 tropicalmed-08-00332-t004:** Time taken to complete the various steps of the care cascade of contact tracing and TPT provision among child household contacts of pulmonary TB patients (index TB patients) initiated on anti-TB treatment in Bishkek, Kyrgyzstan during October 2021 to September 2022.

Time Taken (in Days) between	<5 Years			5–14 Years			Total		
	N ^1^	n ^2^	Median (IQR)	N ^1^	n ^2^	Median (IQR)	N ^1^	n ^2^	Median (IQR)
Treatment initiation of index patient to contact screening	103	75	41 (7–88)	158	125	48 (10–100)	261	200	45 (8–100)
Contact screening to anti-TB treatment initiation	11	11	7 (4–26)	12	12	6 (4–23)	23	23	7 (4–26)
Contact screening to TPT initiation	31	30	4 (3–6)	33	29	4 (3–5)	64	59	4 (3–6)
Treatment initiation of index patient to anti-TB treatment/TPT initiation in child household contact	42	41	21 (3–73)	45	41	25 (9–95)	87	82	24 (6–74)
Number (%) initiated on anti-TB treatment/TPT within 15 days of initiating treatment for index TB patient	42	41	16 (39%)	45	41	13 (32%)	87	82	29 (35%)

^1^ Number of child household contacts who were eligible for calculating duration; ^2^ Number of child household contacts for whom a valid date was available for calculating duration; Note: Mann Whitney U test was used to compare the duration between child household contacts aged < 5 years and those aged 5 to 14 years. There was no statistically significant difference in the time taken to complete various steps between child household contacts aged under five years and those who were 5 to 14 years.

## Data Availability

Requests to access these data should be sent to the corresponding author.

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
