# Peer review of "Contact Tracing and Tuberculosis Preventive Therapy for Household Child Contacts of Pulmonary Tuberculosis Patients in the Kyrgyz Republic: How Well Are We Doing?"

_tropicalmed, 2023, doi:10.3390/tropicalmed8070332_

Round 1

Reviewer 1 Report

“Tracing, Screening and Tuberculosis Preventive Therapy for Household Child Contacts of Pulmonary Tuberculosis Patients in the Kyrgyz Republic: How Well Are We Doing?” is interesting topic.

Although, the scope and originality of the study is better presented throughout the manuscript, but there are few corrections needed in the manuscript.

Here are the main comments that should be considered__

Title

The title, as presented, captures adequately the central theme of the study. However, the authors should rewrite it little for better understanding.

Keywords

Are abbreviations allowed in the keywords? especially the SORT IT, were not defined at first mention.

The introduction, presented, is comprehensive. It addresses the import of this research. The actual problem necessitating the conduct of this research was presented and well-justified. The aim of the study was also presented. However, the introduction must discuss about pathophysiology of bacteria that causing TB, and resistivity mechanisms. 

Line 156 Authors clarify it, without “Antimicrobial susceptibility test” Anti-TB drugs given to children.  

The manuscript showed several results that cannot ignored, but a paper of the high quality must show consistent discussion. Please improve the discussions for the several results topics.

Conclusion:

The conclusion and discussion are repetitive to each other. 

The conclusion does not provide aspects mentioned in the title and body of the text as epidemiological perspectives, and clinical and diagnostic advances, and future plan according to socio-demographic aspects. 

Please mention the plan, which already implemented by State government, and if you see any improvement please discuss and the data. 

Please mention the “Limitations” of this study. Future reassessment of data for risk relationships of several risk factors, such as ambient population, damage resulting from reinfection or resistivity. 

Minor editing of English language required.

Author Response

“Tracing, Screening and Tuberculosis Preventive Therapy for Household Child Contacts of Pulmonary Tuberculosis Patients in the Kyrgyz Republic: How Well Are We Doing?” is interesting topic.

Although, the scope and originality of the study is better presented throughout the manuscript, but there are few corrections needed in the manuscript.

Response: Thank you for the review and constructive comments. We have responded to the comments below and have made necessary changes to improve the quality of the manuscript.

Title

The title, as presented, captures adequately the central theme of the study. However, the authors should rewrite it little for better understanding.

Response: Thank you for the valid suggestion. We have changed the title to remove the multiple single words at the start of the title. Keeping the central theme intact, we have changed the title to “Contact Tracing and Tuberculosis Preventive Therapy for Household Child Contacts of Pulmonary Tuberculosis Patients in the Kyrgyz Republic: How Well Are We Doing?”.

Keywords

Are abbreviations allowed in the keywords? especially the SORT IT, were not defined at first mention.

Response: Thank you for the comment. We agree with the reviewers comment on the use of abbreviation as keywords. We have now changed SORT IT to ‘Structured Operational Research Training IniTiative’.

However, in the past, the Tropical Medicine and Infectious Disease (TMID) journal has allowed the use of SORT IT as the keyword. SORT IT being an operational research training programme of the WHO-TDR, the Special Programme for Tropical Disease Research, we have used SORT IT as a keyword in all the articles through this operational research training programme. In Tropical Medicine and Infectious Disease (TMID), more than 30 articles have been published with the SORT IT as a key words.

The introduction, presented, is comprehensive. It addresses the import of this research. The actual problem necessitating the conduct of this research was presented and well-justified. The aim of the study was also presented. However, the introduction must discuss about pathophysiology of bacteria that causing TB, and resistivity mechanisms. 

Response: Thank you for the comment. The current operational research study largely looks at the programmatic implementation of the contact investigation and provision of TB preventive therapy for children in the Kyrgyz Republic. Thus, we had not detailed on the pathophysiology of the TB as it was not relevant to the current research topic. However, in line with the comment from the learned reviewer, we have now added a paragraph on the pathophysiology of TB in line number 46 to 51.

Line 156 Authors clarify it, without “Antimicrobial susceptibility test” Anti-TB drugs given to children. 

Response: Thank you for this suggestion. The anti-TB treatment regimen in children is started based on the drug-sensitivity pattern. This has been clarified in line number 162-163.

The manuscript showed several results that cannot ignored, but a paper of the high quality must show consistent discussion. Please improve the discussions for the several results topics.

Response: Thank you for the suggestion. We believe that all the results do not warrant a discussion in the research manuscript in contrast to thesis or dissertation. We have summarised the five key findings upfront in the discussion section of the manuscript. Each of these key findings have been discussed in detail in lines 362 to 438, as these are the findings that have significant implications for the programme. We will be happy to attend to any specific comments that the reviewer may have on the discussion.

Conclusion:

The conclusion and discussion are repetitive to each other. The conclusion does not provide aspects mentioned in the title and body of the text as epidemiological perspectives, and clinical and diagnostic advances, and future plan according to socio-demographic aspects. 

Response: Thank you for the suggestion. In the conclusion we have mentioned the key findings in line with the objectives of the study and have mentioned the actionable recommendations in line with the study findings. As this study is an operational research, programmatic perspective is given emphasis over the other academically important aspects.

As the discussion and conclusion sections are expected to be standalone, the repetition of the key findings are inevitable.

Please mention the plan, which already implemented by State government, and if you see any improvement please discuss and the data. 

Response: Thank you for the suggestion. The state government has not implemented any new specific intervention, other than those detailed in the methods and discussion section of the manuscript. The results of the current study design which used index patient data of one year, cannot be used to make any interpretation of improving trends in care provision. Also, the newer interventions like digitization of the TB patient records was not established during the study reference period.

Please mention the “Limitations” of this study. Future reassessment of data for risk relationships of several risk factors, such as ambient population, damage resulting from reinfection or resistivity. 

Response: Thank you for the suggestion. There is a detailed description of limitations of the study under the discussion section in line number 339 to 360. The limitations are mentioned upfront before discussing the key findings, as the findings needs to be interpreted in line with the limitations.

We have highlighted the importance of reassessment through real-time monitoring of contact tracing performance in line number 476 to 478.

Reviewer 2 Report

A sound work  with a clear message

Additional comments: 

This paper deals with the problem of identification, screening and investigation  for tuberculosis among children in contact with patients affected by tuberculosis. This represent an important study aimed at  preventing the spreading of the disease. The study was addressed to populations leaving in Bishkek (Kyrgyz Republic) .  To carry on the study,  data were carefully obtained in the period October 2021-September 2022. The authors recognize as a potential limitation of the study the relatively small sample of child household contacts. This may well  be true, yet the importance of the study within the healthcare system is valid.  

I do not have any specific advice to improve the presentation.

I believe that, based on the available data base to carry on the study, any effort is now  required to continue diligently (a word used by the same authors!) to set a follow up on this important topic.

In summary, I consider this study as an important step in preventive medicine.

References are OK  

Figure 1 clearly shows the “care cascade of contact”.

Author Response

A sound work  with a clear message

Response: Thank you for the appreciation on the project and the manuscript.

Additional comments: 

This paper deals with the problem of identification, screening and investigation  for tuberculosis among children in contact with patients affected by tuberculosis. This represent an important study aimed at  preventing the spreading of the disease. The study was addressed to populations leaving in Bishkek (Kyrgyz Republic) .  To carry on the study,  data were carefully obtained in the period October 2021-September 2022. The authors recognize as a potential limitation of the study the relatively small sample of child household contacts. This may well  be true, yet the importance of the study within the healthcare system is valid.

Response: Thank you for acknowledging the importance of the study in spite of limitations related to the  sample size and data deficiency 

I do not have any specific advice to improve the presentation.

I believe that, based on the available data base to carry on the study, any effort is now  required to continue diligently (a word used by the same authors!) to set a follow up on this important topic.

Response: Thank you for the comment. As suggested, we have highlighted the importance of routine monitoring in line number 422 to 428. 

In summary, I consider this study as an important step in preventive medicine.

References are OK  

Figure 1 clearly shows the “care cascade of contact”.

Response: Thank you for the appreciation. 

Round 2

Reviewer 1 Report

I appreciate the authors, they did improve the manuscript. Now I recommend this revised manuscript for publication.